# Structural basis for mechanotransduction in a potassium-dependent mechanosensitive ion channel

Jonathan Mount [1,2], Grigory Maksaev [1,2], Brock T. Summers[3], James A. J. Fitzpatrick [1,3,4,5] & Peng Yuan [1,2] ✉

Mechanosensitive channels of small conductance, found in many living organisms, open under elevated membrane tension and thus play crucial roles in biological response to mechanical stress. Amongst these channels, MscK is unique in that its activation also requires external potassium ions. To better understand this dual gating mechanism by force and ligand, we elucidate distinct structures of MscK along the gating cycle using cryo-electron microscopy. The heptameric channel comprises three layers: a cytoplasmic domain, a periplasmic gating ring, and a markedly curved transmembrane domain that flattens and expands upon channel opening, which is accompanied by dilation of the periplasmic ring. Furthermore, our results support a potentially unifying mechanotransduction mechanism in ion channels depicted as flattening and expansion of the transmembrane domain.

Mechanosensitive (MS) ion channels are central for all living organisms to perceive and respond to mechanical stimuli for survival, growth, and development[1–8]. *E. coli* encodes multiple distinct MS channels, of which MS channels of small (MscS) and large conductance (MscL) have been extensively studied since their molecular identification and have served as the prototypical model systems for understanding the biophysical principles of mechanotransduction in ion channel proteins[9–19]. Intrinsically activated by increased lateral membrane tension, MscS and MscL open under hypoosmotic shock, allowing osmolyte efflux and thereby maintaining the integrity of bacterial cells under osmotic challenge[9,10,19]. X-ray and cryo-electron microscopy (cryo-EM) structures of *E. coli* MscS (*Ec*MscS) have revealed the architecture of a heptameric channel, with each subunit containing three membrane-spanning helices and a C-terminal soluble domain[18,20–25]. However, even in light of a wealth of molecular structures, the mechanism by which *Ec*MscS senses membrane tension and transitions to an open, conductive conformation remains incompletely understood and intensely debated[6,15,22,25–27]. This is partially due to the technical challenge of applying mechanical force,

the physiological stimulus that activates MS channels, in high-resolution structural approaches such as X-ray crystallography and single-particle cryo-EM. Thus, to obtain molecular structures representing distinct functional states during the gating of MS channels, especially the open conformations, is immensely difficult but necessary to illuminate the biophysical principles underlying mechanotransduction.

A large family of *Ec*MscS-like channels, which share a common channel core consisting of three transmembrane (TM) helices followed by a characteristic C-terminal soluble domain, has been identified in many lineages encompassing bacteria, protists, fungi, and plants (Fig. 1a)[1,3,28]. Remarkably, these channels have distinct membrane topologies and domain organizations and, consequently, versatile physiological functions in response to mechanical cues[3,29–31]. Therefore, these structurally diverse homologs may present a rare opportunity to examine both common and unique mechanisms governing mechanotransduction. Amongst these family members, MscK is unique in that channel activation, in addition to elevated lateral membrane tension, requires the presence of certain external ions such

[1]Department of Cell Biology and Physiology, Washington University School of Medicine, Saint Louis, MO, USA. [2]Center for the Investigation of Membrane Excitability Diseases, Washington University School of Medicine, Saint Louis, MO, USA. [3]Washington University Center for Cellular Imaging, Washington University School of Medicine, Saint Louis, MO, USA. [4]Department of Neuroscience, Washington University School of Medicine, Saint Louis, MO, USA. [5]Department of Biomedical Engineering, Washington University in Saint Louis, Saint Louis, MO, USA. ✉e-mail: yuanp@wustl.edu

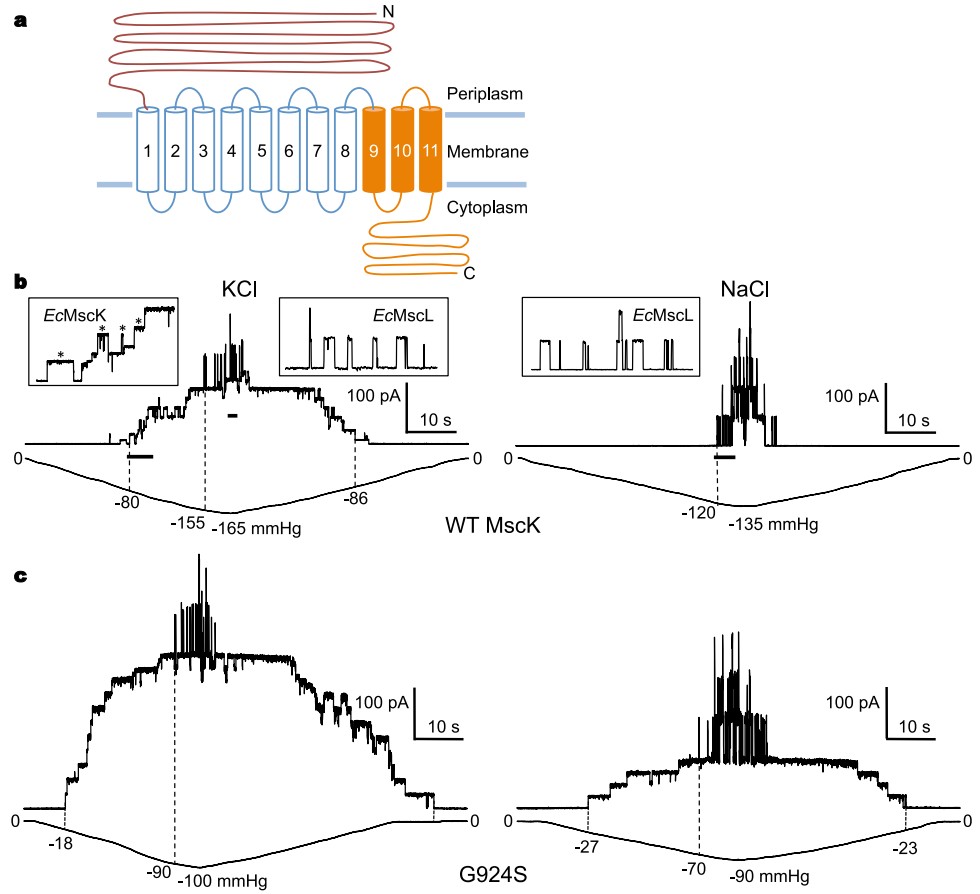

**Fig. 1 | K$^+$-dependent mechanosensitive channel MscK. a** Domain topology of the MscK channel. The *Ec*MscS-like core, shared by all MscS homologs, is colored in orange. In comparison with *Ec*MscS consisting of three TM helices, MscK has a large N-terminal periplasmic domain and eleven TM helices. **b** Current traces from membrane patches of giant spheroplasts from the *E. coli* strain pB113 expressing *Ec*MscK, in 200 mM KCl (left) or 200 mM NaCl (right). Membrane potential was −30 mV. Insets indicate the activity of *Ec*MscK (unitary conductance of 895 ± 25 pS, 15 independent patches, data were mean ± SEM) or *Ec*MscL (unitary conductance of

3.65 ± 0.07 nS, ten independent patches). The symbol '*' indicates MscK activity. MscK activation was observed in symmetric K$^+$, but not Na$^+$, recording solutions with increased negative pressure in the recording pipettes. The endogenous *Ec*MscL channels, which open at high membrane tension, provide calibration of tension in excised membrane patches. **c** Recordings of the *Ec*MscK mutant G924S in 200 mM KCl (left, unitary conductance of 946 ± 19 pS, 11 independent patches) or 200 mM NaCl (right, unitary conductance of 847 ± 12 pS, seven independent patches).

as K$^+$, Rb$^+$, or Cs$^+$ [32–34]. Of note, MscK is the largest known bacterial MS channel in molecular weight. The *E. coli* MscK (*Ec*MscK) channel, consisting of 1120 amino acids, contains a large N-terminal periplasmic domain (PD, ~500 amino acids), a transmembrane domain (TMD) with eleven predicted TM helices, and a cytoplasmic domain (CTD) analogous to that of *Ec*MscS (Fig. 1a) [32–34]. *Ec*MscK is expressed at a much lower density and provides no substantial protection against hypoosmotic shock in cell survival assays despite opening at a much lower membrane tension threshold than *Ec*MscS [10,35]. Electrophysiological recordings from giant *E. coli* spheroplasts have revealed that *Ec*MscK is sensitive to K$^+$ ions only on the periplasmic side, suggesting that ionic regulation is likely conferred by the unique N-terminal PD that precedes the TMD [32]. Supporting this notion, truncated *Ec*MscK constructs containing only the *Ec*MscS homologous region of the channel, but lacking the PD, could be activated by membrane tension alone [36]. Notably, electrophysiological studies have identified several gain-of-function (GOF) point mutations, including natural variants, which have a lower activation tension threshold and/or abolish the ionic requirement [32,33].

To better understand the biophysical underpinnings of mechanotransduction and to establish the molecular basis of dual activation in a ligand- and force-gated channel, we have determined cryo-EM structures of *Ec*MscK in distinct functional states by leveraging GOF mutants. Based on the ion-conduction pore dimensions, we assign

these structures to closed, intermediate, and open conformations of the channel during its gating cycle. Together with electrophysiological studies using *E. coli* spheroplasts, our results illuminate the structural mechanism of MscK gating depicted as the "flattening and expansion" of a curved TMD driven by lateral membrane tension. This mode of mechanotransduction likely applies to numerous MS channels that are characterized by an inherently non-planar TMD embedded in a cell membrane.

## Results

### Function of MscK and structure determination

We examined *Ec*MscK channel activity in excised membrane patches of giant spheroplasts from the *E. coli* strain pB113 [10,32], which lacks endogenous *mscK* and *mscS* genes (Fig. 1b). Endogenous *E. coli* MscL (*Ec*MscL) channels, which open at essentially the same high tension threshold in Na$^+$ or K$^+$ conditions [19,32], provide calibration of tension in excised membrane patches. Consistent with previous studies [32,33], the application of pressure activates *Ec*MscK channels in symmetric KCl, but not symmetric NaCl, recording solutions (Fig. 1b). We expressed the full-length wild-type (WT) *Ec*MscK channel in yeast *P. pastoris* and purified the channel protein to homogeneity for structural studies. Intriguingly, *Ec*MscK could be purified in detergent buffers containing 150 mM NaCl but was prone to aggregation in 150 mM KCl, indicating a direct influence of K$^+$ on the purified channel protein. We

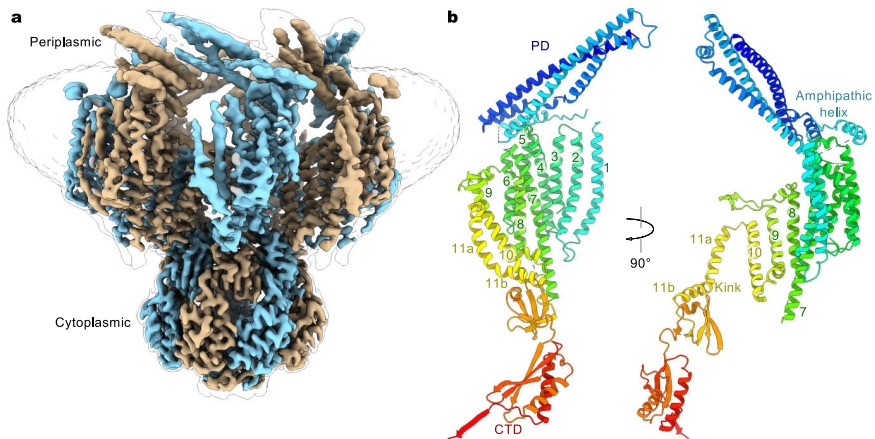

**Fig. 2 | Cryo-EM structure of *Ec*MscK. a** Cryo-EM reconstruction of *Ec*MscK (G924S) in the closed state. Detergent micelle densities are also indicated using the unsharpened map. Channel subunits are in alternate colors. **b** Orthogonal views of a single subunit, showing the arrangement of the periplasmic domain (PD), cytoplasmic domain (CTD), and transmembrane domain (TMD) with 11 TM helices.

conducted single-particle cryo-EM analysis of *Ec*MscK purified with 150 mM NaCl and obtained a nominal resolution of 4.5 Å, as assessed by the Fourier Shell Correlation (FSC) threshold of 0.143 (Supplementary Fig. 1). The limited resolution is likely owing to intrinsic domain motions between the CTD, TMD, and PD, revealed by 3D variability analysis (3DVA) in cryoSPARC (Supplementary Movie 1)[37,38]. To potentially improve structural homogeneity and to achieve a functionally distinct conformation, we decided to characterize a GOF mutant G924S, which, in contrast to the WT channel, was active under elevated membrane tension in both KCl and NaCl conditions (Fig. 1c)[33]. Analysis of single-channel patch-clamp data demonstrated that the G924S mutant retained the unitary conductance of the WT channel (Supplementary Fig. 2). Notably, G924S in NaCl displayed a pressure activation threshold similar to that of the wildtype in KCl, but in KCl opened at a lower threshold (Supplementary Fig. 2). Thus, we purified the *Ec*MscK G924S mutant channel in detergent micelles with either NaCl or KCl and collected single-particle cryo-EM data (Supplementary Fig. 3). Notably, both NaCl and KCl conditions yielded apparently distinct conformations that presumably represent the closed and open states, which we will elaborate further in this article. Independent processing of these two datasets resulted in similar reconstructions and particle distributions in the closed and open states. Thus, the cryo-EM datasets of G924S in NaCl and in KCl were essentially indistinguishable, which allowed us to combine the datasets to further improve the reconstructions of both the closed and open conformations to overall resolutions of 3.84 and 3.47 Å, respectively (Supplementary Fig. 3 and Supplementary Table 1).

In the G924S closed conformation, the channel core, including TM7-11 and the cytoplasmic CTD, had a higher local resolution, which allowed placement of the majority of the side chains (Fig. 2a and Supplementary Fig. 4). The peripheral TM helices (TM1-6) and the periplasmic PD were less well resolved, and thus most side chains were not modeled. In the open conformation, the density for TM9-11 and the CTD is sufficient for building an atomic model, while TM1-8 and the PD could be well modeled by nearly rigid-body adjustment of the corresponding region from the closed conformation. The final atomic models fit well into their respective densities and were refined to good stereochemistry (Supplementary Fig. 4 and Supplementary Table 1). The lower resolution reconstruction of the WT channel is nearly identical to the closed conformation of G924S (Supplementary Fig. 5), and hereafter the higher-resolution structures of G924S are primarily used for structural analysis and interpretation.

## Channel architecture

The MscK channel forms a symmetric heptamer, with each subunit comprised of three distinct layers: an N-terminal periplasmic PD, a TMD with eleven TM helices, and a C-terminal cytoplasmic CTD (Fig. 2a). In the periplasm, the resolved C-terminal portion of each PD forms a helix bundle, and the seven PDs assemble into a ring structure above the membrane. The PD is connected to the TMD via a short peripheral amphipathic helix (Fig. 2a, b and Supplementary Fig. 6), which is presumably located at the boundary of the periplasm and the inner membrane and defines the outer perimeter of the transmembrane region of the channel (Fig. 2a). Of note, an amphipathic helix at the membrane boundary, which is often found in structurally unrelated MS channels such as MscL, TRAAK, OSCA, and PIEZOs, has been thought to function as a critical structural element coupling membrane dynamics to channel conformation[39–42]. Thus, the horizontal helix in MscK may serve an analogous role in sensing membrane tension but also directly coordinate structural rearrangements in the periplasmic and transmembrane domains by providing a covalent linkage.

Following the amphipathic helix is a large TMD with eleven membrane-spanning helices (Fig. 2b). The innermost pore-lining helix TM11 contains an apparent kink such that TM11a defines the transmembrane ion-conduction path and TM11b runs nearly parallel to the membrane plane and extends to the cytoplasmic CTD (Fig. 2b). The overall structure of the channel core, including the arrangement of inner TM helices (TM7-11) and the cytoplasmic CTD, resembles those of the extensively studied prokaryotic MscS homologs and the recently characterized eukaryotic MSL1 channels (Supplementary Figs. 6, 7)[18,20–25,43,44]. Common structural features include a kinked pore-lining helix, arrangement of additional TM helices in a virtually straight line, and seven cytoplasmic side portals formed by the CTDs, which presumably allow ion passage and contribute to ion selectivity and unitary conductance, as demonstrated in *Ec*MscS and *At*MSL1[44,45]. Notably, while the conserved CTDs align well with each other, the TM helices are distinctly arranged in the *Ec*MscK channel (Supplementary Fig. 7f, g). In comparison with *Ec*MscS and *At*MSL1, the TMD in *Ec*MscK rotates counterclockwise, viewed from the periplasmic or extracellular side, which pivots at the helical kink.

## Curved transmembrane domain

The eleven TM helices in the TMD of MscK can be divided into two groups, with the outer TM1-6 forming a peripheral helix bundle and the inner TM7-11 constituting the channel core together with the CTD, which accounts for major inter-subunit interactions in the

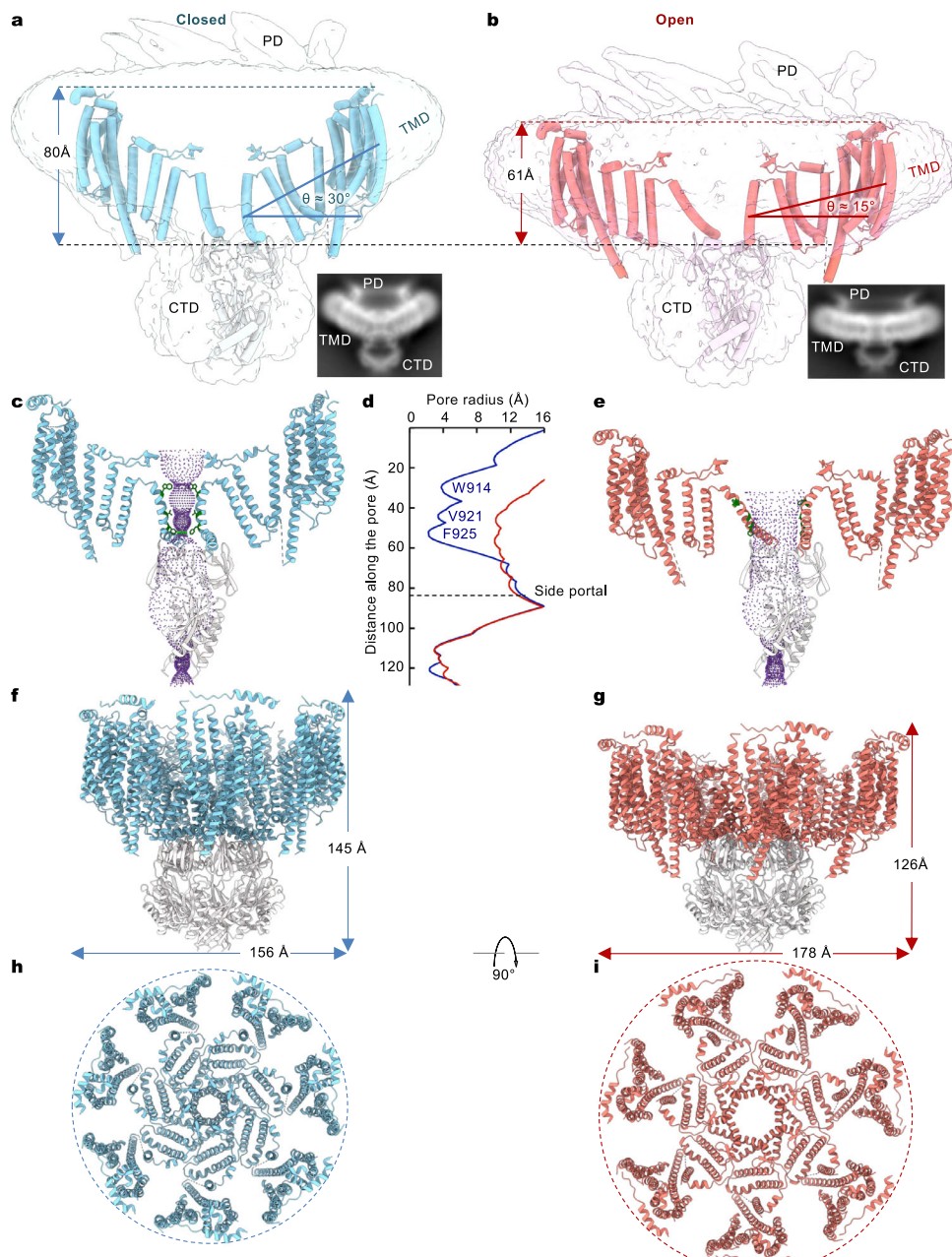

**Fig. 3 | Opening of *Ec*MscK. a**, **b** Closed (**a**) and open (**b**) structures of *Ec*MscK G924S. Only two opposing subunits are shown for clarity. The surfaces of the unsharpened maps indicate the positions of the PD, TMD, and CTD. Dotted lines indicate the cytoplasmic and periplasmic boundaries. Insets are 2D class averages representing side views of the channel (Fig. 3a). illustrating the curvatures of the TMD. **c** The closed pore. **d** Comparison of the pore dimensions in the closed and open states. Pore-lining residues defining the gates are highlighted. **e** The open pore. **f**, **g** Dimensions of the channel in the closed (**f**) and open (**g**) conformations. **h**, **i** The TMD in the closed (**h**) and open (**i**) states.

heptameric channel assembly (Fig. 2 and Supplementary Fig. 7c). In accordance with this structural arrangement, the channel core was better resolved in our cryo-EM reconstructions than the peripheral TM1-6 bundle, which appears to be structurally dynamic (Supplementary Fig. 3). Analogous to *At*MSL1, the heptameric MscK channel has limited inter-subunit packing interactions within the membrane, leaving large unoccupied spaces between neighboring subunits that are presumably filled with lipids in the bacterial inner membrane (Fig. 2a and Supplementary Fig. 7)[43]. The substantial protein-free region in the TMD supports the notion that MS channels, in general, are less densely packed than other membrane channels or transporters[46].

The transmembrane region of MscK is discernibly bent, which is also evident by the reference-free 2D class averages of particles representing side views of the channel (Fig. 3a). In cell membranes, a curved TMD would induce deformation of the surrounding lipid bilayer, which is otherwise planar without perturbation. While it has previously been speculated that the extended TMD of MscK may influence its localization to the curved poles of an *E. coli* cell, photo-activated localization microscopy measurements in native cells have ruled out this possibility[47]. The arrangement of the eleven TM helices generates a midplane bending angle of ~30° (Fig. 3a). The unusual non-planar shape of the TMD is reminiscent of the structurally related plant MSL1 as well as unrelated animal PIEZO channels[41,43,48,49]. Thus, it is tempting to speculate that these structurally and functionally distinct channels may share a common gating mechanism rooted in membrane deformation at the boundary of the embedded channel and its surrounding lipid bilayer.

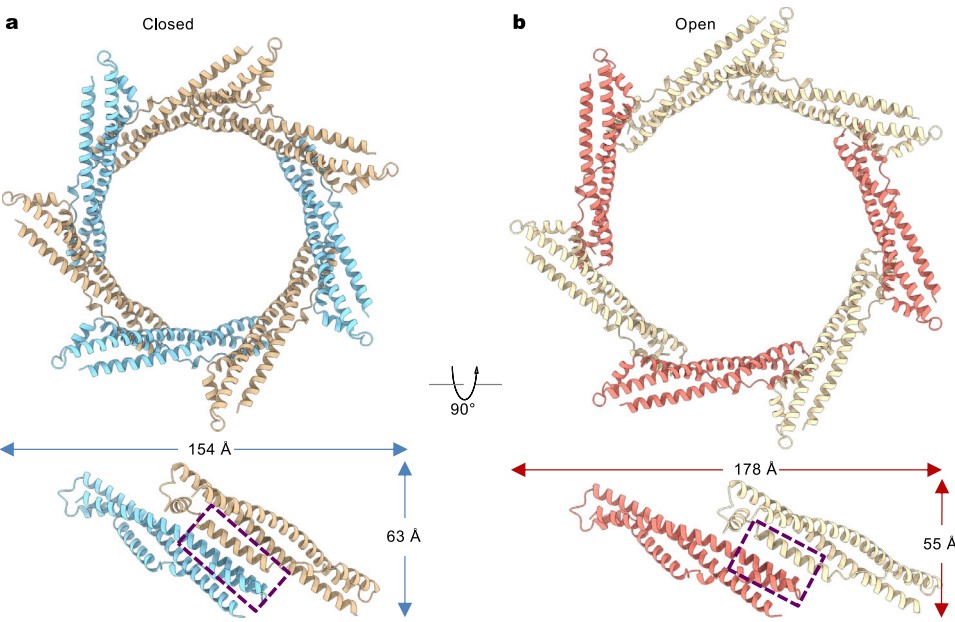

**Fig. 4 | Periplasmic gating ring. a, b** Structures of the PD in the closed (**a**) and open (**b**) states. The overall dimensions and the helix-packing interactions at the inter-subunit interfaces are indicated.

## Opening of MscK

Cryo-EM analysis of the GOF mutant G924S revealed an open conformation indicated by pore radius estimation (Fig. 3b–i and Supplementary Fig. 3). Akin to *Ec*MscS and *At*MSL1, the pore-lining helix (TM11) in the resting state is kinked, and several bulky hydrophobic side chains point to the central pore axis to create narrow constrictions (Fig. 3c, d and Supplementary Fig. 6). A hydrophobic gate near the cytoplasm is constituted by V921 and F925, which are conserved amongst MscK homologs. Notably, corresponding residues in the pore-lining helix in *Ec*MscS (L105 and L109) or *At*MSL1 (V319 and F323) form an analogous hydrophobic gate that widens upon channel activation (Supplementary Fig. 6)[20,43]. The pore diameter of the cytoplasmic gate (~5 Å) in the closed MscK conformation is comparable to that of *Ec*MscS (~5 Å) and smaller than that of *At*MSL1 (~8 Å). In contrast to *Ec*MscS and *At*MSL1, an additional constriction at the periplasmic side of MscK, with a pore diameter of ~8 Å, is formed by a ring of seven tryptophan (W914) residues (Fig. 3c, d and Supplementary Fig. 6). It appears that MscK possesses two physically separated hydrophobic gates along the pore: an upper periplasmic gate and a lower cytoplasmic gate. In the open conformation of MscK, the pore-lining helix TM11 straightens and would be more tilted within the membrane. The bulky side chains of gate residues, W914, V921, and F925, rotate to the side and move further away from the central pore axis (Fig. 3c–e). The combination of helix straightening and side chain movement gives rise to a substantially enlarged pore with a minimal diameter of ~20 Å within the TM region, which is comparable to the open pore of *At*MSL1 (Fig. 3d, e)[43].

The lower cytoplasmic constriction represents a common gate shared by *Ec*MscK, *Ec*MscS, and *At*MSL1[18,20,43]. However, the upper periplasmic constriction, generated by a conserved tryptophan residue (W914), is a unique structural feature in MscK. To correlate this structural finding, we introduced a point mutation W914A and found that the mutant channel had a wild-type unitary conductance and was activated at a lower tension threshold than the wildtype (Supplementary Figs. 2, 8). Furthermore, the mutant channel was characterized by frequent transitions between the closed and open states. It appears that the elimination of the bulky side chain destabilizes the open conformation and that W914 is indeed involved in channel gating. Presumably, the W914A mutation decreases the free energy barrier associated with the flipping of the bulky tryptophan side chain during the gating transition

(Supplementary Fig. 8), rendering a mutant channel that can more readily alternate between the open and closed states.

Transition to the open state involves substantial structural rearrangements of both the TMD and the periplasmic gating ring. In contrast, the cytoplasmic domain remains essentially unaltered (Fig. 3c–i and Supplementary Movie 2). Strikingly, the PD and the majority of the TMD, except for the pore-lining helix, virtually move outward together as a single structural unit, giving rise to a nearly flat transmembrane region in the open state (Supplementary Movie 2). Flattening of the TMD is also apparent by comparison of model-free 2D class averages of the closed and open conformations (Fig. 3a, b). Upon channel opening, the midplane bending angle in the membrane is reduced from ~30° to ~15°. The global conformational change in the channel results in a reduction of 19 Å in height and an expansion of 22 Å of the TMD in the membrane plane (Fig. 3f–i). The downward and outward rigid-body movement of the PD-TMD entity pulls open the pore-lining helix TM11a, which then joins TM11b via the straightening of the kink at G926. Consequently, the collective movement of the seven pore-lining helices, in combination with side chain rotations of pore-narrowing residues, generates a wide-open transmembrane pore.

The periplasmic gating ring is primarily held together by packing interactions of inter-subunit three-helix bundles, in which one of the interfacial helices engages its entirety at the interface in the closed conformation (Fig. 4a). Accompanying channel opening, the ring dilates from 154 to 178 Å, and the packing interactions are likely weakened at the assembly interface as the engagement of the three helices in the bundles is reduced (Fig. 4b). Expansion of the periplasmic ring appears to be obligatory upon channel opening due to direct attachment to the TMD via the amphipathic helix (Fig. 2). As external ions such as K⁺ regulate channel activity from the periplasmic side, it is plausible that these ions directly bind to the ring and influence its conformation, which is subsequently transmitted to the TMD through the bridging amphipathic helix.

The nature of the gating transition in *Ec*MscK is reminiscent of that in *At*MSL1[43]. In both channels, opening involves flattening and expansion of the TMD, which would be intrinsically curved in the resting state and induce membrane deformation surrounding the embedded channel[43]. Analogous conformational changes occur in both channels. First, the soluble CTD essentially maintains a static

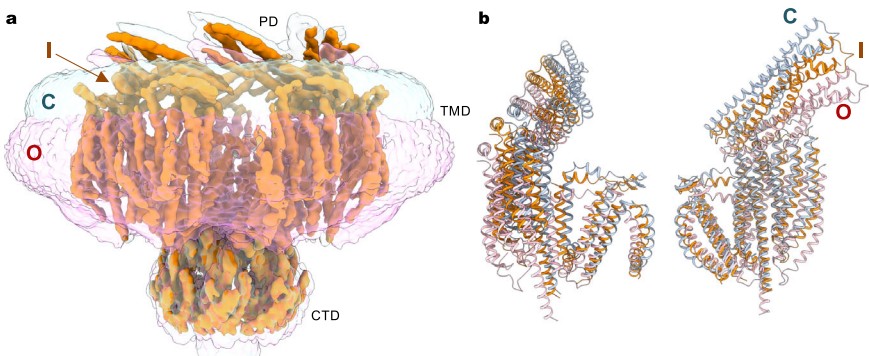

**Fig. 5 | Intermediate state. a** Reconstruction of an intermediate conformation (I, orange), in comparison with the closed (C, light blue) and open (O, light red) conformations. The sharpened density map of the intermediate state is aligned with the unsharpened closed and open density maps, which are shown at low contour levels to highlight the shapes. **b** Superposition of the closed (C), intermediate (I), and open (O) structures.

structural scaffold during gating while the TMD undergoes drastic structural rearrangement characterized by nearly rigid-body movement. Second, the kinked pore-lining helices in the closed states become straightened and more tilted within the membrane upon opening[43]. Meanwhile, considerable distinctions are also present in these two gating transitions. The degree of curvature of the TMD in the resting state and the nature of the rigid-body movement of the TMD are markedly different. *At*MSL1 displays a sharply bent, bowl-shaped TMD, which undergoes both translational and rotational rearrangements to reach a flat and expanded transmembrane region[43]. In contrast, *Ec*MscK has a less curved TMD with a midplane bending angle of ~30°, which simply moves outward and downward to reduce the bending angle and dilate the central pore without additional rotation. The distinct TMD rearrangement of MscK during gating may originate from its unique periplasmic ring structure immediately above the membrane, in which each PD is covalently connected to the TMD via a peripheral horizontal helix. Thus, the heptameric periplasmic ring, which also dilates upon channel opening, would, in principle, restrain the extent of conformational changes permitted in the TMD.

### Intermediate conformation

An intermediate conformation, distinct from both the closed and open structures, was revealed by 3D classification and refined to a nominal resolution of 5.0 Å (Fig. 5a and Supplementary Fig. 3). The CTD, TMD, and PD are resolved in the density map, albeit with limited resolution. Comparison with the closed and open structures indicates that this may represent an intermediate conformation advancing to channel activation (Fig. 5a, b). We built an atomic model by rigid-body adjustment of the PD and most of the TMD from the closed structure, and the resulting model fit well into the density. The pore dimensions could not be accurately calculated because the limited resolution prevented the placement of side chain conformations. Therefore, we restrained our interpretation at the level of global conformational changes. The transition from the closed to the intermediate state involves structural rearrangements of the PD and TMD analogous to those in full channel opening, but to a lesser extent (Fig. 5b and Supplementary Movie 3). Thus, the intermediate structure most likely represents a conformational state along the pathway to activation and further supports the notion that pore opening is largely driven by a nearly rigid-body movement of the PD-TMD unit.

### Discussion

Mechanosensation underlies our basic senses of hearing and touch and may represent the most ancient sense in early life, as it would be critical to the survival of single-cellular organisms living in a dynamic osmotic environment. However, our understanding of the underlying physicochemical principles of mechanoreceptors remains rather limited and falls far behind our understanding of other senses, such as smell and taste, which are typically elicited by specific ligand-receptor interactions that are more amenable to detailed biochemical and structural analyses. How MS channels perceive mechanical force and transition to conductive states remains a central question that is yet to be addressed.

This work illuminates the opening of MscK, an MS channel that is additionally regulated by ligand. We infer that tension drives channel opening through flattening and expansion of the TMD, which is curved in the closed conformation and induces a midplane bending angle ($\theta_c$) of ~30° ($\theta_c = \pi/6$) in the surrounding lipid membrane. In comparison with the closed conformation, the open conformation has a reduced bending angle ($\Delta\theta = 15°$) and an expanded in-plane cross-sectional area ($\Delta A = 2\pi R_c * \Delta R = 54$ nm²), the free energy difference between the curved and flattened conformations is determined by

$$\triangle G = \triangle G_{\sigma=0} + \triangle G_{\text{bending}} - \sigma \triangle A \qquad (1)$$

where $\triangle G_{\sigma=0}$ is the difference of free energy intrinsic to channel conformations at zero membrane tension. Thus, an open conformation of MscK with a much larger cross-sectional area would be favored under elevated membrane tension. The difference in free energy of midplane bending between the closed (in-plane radius $R_c$, midplane bending angle $\theta_c$) and open (in-plane radius $R_o$, midplane bending angle $\theta_o$) states can be approximated by

$$\Delta G_{\text{bending}} = \pi \left( \theta_c^2 R_c - \theta_o^2 R_o \right) \sqrt{\sigma K_b} \qquad (2)$$

where $K_b$ is the bilayer bending modulus ~20 $k_B T$ ($k_B$ is Boltzmann's constant, and $T$ is the temperature)[50]. The contribution of membrane deformation to tension sensitivity can be estimated by the midpoint membrane tension ($\sigma_{1/2}$), at which the open probability of the channel is 0.5. Without consideration of $\triangle G_{\sigma=0}$, $\sigma_{1/2}$ is determined by

$$\pi(\theta_c^2 R_c - \theta_o^2 R_o)\sqrt{\sigma_{1/2} K_b} - \sigma_{1/2} \Delta A = 0 \qquad (3)$$

Thus, the midpoint tension is calculated by

$$\sigma_{1/2} = \frac{(\theta_c^2 R_c - \theta_o^2 R_o)^2 K_b}{4 R_c^2 \Delta R^2} \qquad (4)$$

In MscK, the parameters deduced from the closed and open structures result in an estimated midpoint membrane tension of 0.16 $k_B$T/nm², much lower than the reported midpoint tension of MscL (2.5 $k_B$T/nm²)[51], which is primarily determined by the energetics of

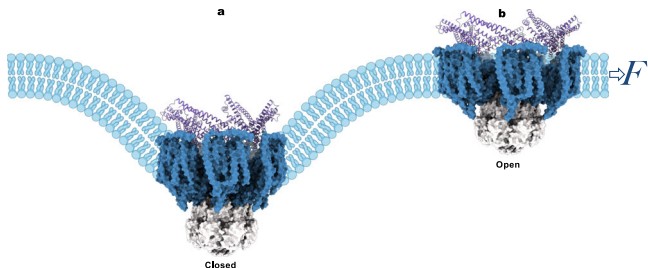

**Fig. 6 | Model for MscK gating. a, b** MscK in the closed, resting (**a**) and open, conductive (**b**) states depicted in a lipid bilayer. Membrane curvature induced by the TMD is indicated (**a**). Membrane tension drives the flattening and expansion of the TMD necessary for the opening of the transmembrane pore. This gating process is additionally regulated by the periplasmic gating ring, which may sense external ionic conditions. The cytoplasmic (gray), transmembrane (blue), and periplasmic (purple) domains are individually colored.

hydrophobic mismatch, rather than midplane bending, between the TM helices and the surrounding lipid bilayer. Opening of MscK is essentially achieved by rigid-body movement of the TMD, which appears to maintain a constant membrane thickness matching that of the bacterial inner membrane. Thus, the contribution of hydrophobic mismatch, which may be involved in channel gating, is not included in our estimation of the tension sensitivity of MscK. This analysis implies that MscK could display much higher tension sensitivity than MscL. Analogously, in PIEZO channels, the extendedly curved TMD induces a curved 'membrane footprint' that amplifies tension sensitivity[52]. In our analysis, the contribution of the intrinsic free energy difference between the closed and open conformations, $\triangle G_{\sigma=0}$, is not included. Notably, the closed and open MscK structures suggest that additional free energy cost must be paid to expand the periplasmic gating ring, thus a higher midpoint membrane tension would be expected.

Achievement of the open conformation of MscK was made possible by the introduction of a single mutation (G924S), which increased tension sensitivity and abolished the $K^+$ requirement. Distinct conformations representing the closed, intermediate, and open states and a similar distribution of these conformational states were obtained in both the $Na^+$ and $K^+$ conditions in cryo-EM experiments. This contrasts with channel activities observed in excised membrane patches, in which the mutant channel, G924S, remained closed in the absence of applied pressure to the excised patches and displayed a lower gating pressure threshold in $K^+$, than $Na^+$, recording conditions. These discrepancies likely originate from detergent micelles, which are necessary to isolate channel protein for cryo-EM but drastically differ from the excised cell membranes in electrophysiology experiments. Notably, the detergent environment, devoid of lipids, appears to additionally modify the gating properties and does not necessarily recapitulate channel properties in the lipid bilayer, as has been observed in MscS[24]. In this work, detergent-solubilized MscK channels, in the absence of tension, have provided unanticipated insights into channel gating and allowed us to infer the mechanotransduction process in an ion channel.

Remarkably, single amino acid substitutions between the two consecutive pore-narrowing residues in *Ec*MscS (A106V), *At*MSL1 (A320V), and now *Ec*MscK (G924S) all resulted in open conformations (Supplementary Figs. 6, 9)[20,43]. These observations support the notion that the pore-lining helix in various MscS homologs is enriched in small-sized glycine and alanine residues, which enable tight helix-helix associations critical for stabilization of the closed conformation[20,43]. Consequently, the introduction of a bulkier side chain at one of these locations can promote channel opening, likely through destabilization of the closed conformation. Notably, another naturally occurring mutation in this region, G922S in *Ec*MscK, also rendered the mutant channel more sensitive to tension[32,33]. These data and analyses further indicate shared physicochemical principles in the gating of these channels.

Though MscK and MSL1 have distinct domain organization and activation modality, the opening of both channels is characterized by drastic "flattening and expansion" of an inherently curved TMD in the closed state that induces membrane deformation in its surrounding lipid bilayer (Fig. 6)[43]. The common biophysical properties of these two channels suggest a potentially unifying mechanical gating mechanism originated from a non-planar TMD, which is further supported by structural and HS-AFM studies of unrelated PIEZO channels[46,48,53]. The distinct molecular structures of various MS channels may dictate specific gating behavior, but in essence, membrane deformation induced by the non-planar shape of the TMD may underlie a universal mechanotransduction process in channels containing this structural feature. The degree of curvature intrinsic to these channels and, subsequently, the extent of membrane deformation would vary and may determine tension sensitivity and precise structural rearrangement leading to channel opening. For instance, extra domain structures such as the periplasmic gating ring in MscK could confer additional regulation mechanisms and further modulate tension sensitivity (Fig. 6). Nonetheless, the underlying biophysical principles governing mechanotransduction are likely shared by many MS channels. Certainly, delineation of the gating transitions of additional MS channels, especially for those with distinct architectures, would provide further assessment of this seemingly unifying mechanical gating mechanism depicted as force-induced flattening and expansion.

## Methods
### Cloning
DNA encoding *E. coli* MscK (*Ec*MscK) was codon-optimized and synthesized into the pUC57 plasmid (Bio Basic, Inc), and served as the template for subsequent cloning. The DNA fragment corresponding to full-length *Ec*MscK was isolated using the restriction enzymes EcoRI and XhoI and subsequently ligated into a modified yeast *P. pastoris* expression vector pPICZ-B that contains a C-terminal PreScission protease cleavage site and a GFP-His₁₀ tag. For expression in *E. coli* spheroplasts, the *Ec*MscK DNA fragment was digested from the pPICZ-B vector and gel purified, and then ligated into PCR-amplified pET300 vector by Gibson cloning. The G924S (primers 5′-TCATTTGGATTGC AAGAAATTTTCGGCAACTTCGTGTCTGG-3′ and 5′-CAATCCGACGGAC AGGGCAGCAGCAAGCCA-3′) and W914A (primers 5′-CTGCAACTTA TCCCAAGACACGCCCAATGATCCAAAGAC-3′ and 5′-GCGCTTGCTGC TGCCCTGTCC-3′) mutations were introduced by site-directed mutagenesis.

### Expression and purification
The wild-type *Ec*MscK and G924S mutants were expressed in *P. pastoris* (strain SMD1163H, Invitrogen). Yeast cells were harvested, and 80 g of cells were disrupted by milling (Retsch MM400) and resuspended in buffer A containing 50 mM Tris-HCl pH 8.0, 150 mM NaCl, and a mixture of protease inhibitors (2.5 μg ml⁻¹ leupeptin (L-010-100, GoldBio), 1 μg ml⁻¹ pepstatin A (P-020-100, GoldBio), 100 μg ml⁻¹ 4-(2-Aminoethyl) benzenesulfonyl fluoride hydrochloride (A-540-10, GoldBio), 3 μg ml⁻¹ aprotinin (A-655-100, GoldBio), 1 mM benzamidine (B-050-100, Gold-Bio), and 200 μM phenylmethane sulphonylfluoride (P-470-25, Gold-Bio) and DNase I (D-300-1, GoldBio) to a final volume of 240 ml. For the G924S mutant, the resuspension was stirred at 4 °C for 3 h, followed by centrifugation for 20 min at 2500 × *g*, 4 °C to remove the insoluble cell fraction. The supernatant was retained and subjected to ultra-centrifugation at 4 °C, 100,000 × *g* for 1 h. The new supernatant was discarded, and the pellet was rinsed with a resuspension solution containing 150 mM NaCl, 20 mM Tris-HCl pH 8.0, and a protease inhibitor mixture. The pellet was divided into four equal aliquots, and each was vortexed into 40 ml of resuspension solution followed by Dounce homogenization on ice. Glyco-diosgenin (GDN, GDN101, Anatrace) was added to the mixture at 1% w/v and rotated at 4 °C for 3 h. The

solubilized fraction was incubated with 2.5 ml of cobalt-charged resin (786-403, G-Biosciences) per 40 ml of pichia membrane solution for 4 h at 4 °C with rotation. Each 2.5 ml of cobalt-charged resin was separated and washed with 10 column volumes of wash solution containing 20 mM Tris-HCl pH 8.0, 150 mM NaCl, 0.02% GDN and 30 mM imidazole, and eluted with 10 ml of this wash solution with the addition of 300 mM imidazole. PreScission protease was added to the elution to remove the GFP-His$_{10}$ tag at 4 °C for 6 hours with gentle rocking. After cleavage, the protein was concentrated using an Amicon Ultra concentrator with a 100 kDa molecular weight cutoff and further purified on a Superose 6 increase 10/300 column (GE Healthcare Life Sciences) equilibrated with a buffer containing 20 mM Tris-HCl pH 8.0, 150 mM NaCl and 0.02% GDN. The peak fractions were combined and concentrated to ~7 mg ml$^{-1}$ for cryo-EM experiments. For channel purification in KCl, the procedures were the same, except that NaCl was replaced by KCl in all buffers. The wild-type channel was purified in the same manner as the mutant channel with the exception of the initial extraction step, where the channel was extracted from the whole cell lysate by solubilization with Lauryl Maltose Neopentyl Glycol (LMNG, NG310, Anatrace) to a final concentration of 1% w/v while stirring at 4 °C for 3 h. After extraction, the channel protein was exchanged into 0.02% GDN on resin, and further purified as described above.

## Cryo-EM sample preparation and imaging

Cryo-EM grids were prepared using FEI Vitrobot Mark IV (FEI). A volume of 3 µl of purified channel protein (~7 mg ml$^{-1}$) was applied. Samples of *Ec*MscK G924S in NaCl and KCl were applied onto glow-discharged copper Quantifoil R1.2/1.3 (Q350CR1.3, Electron Microscopy Sciences) and R2/2 (Q350CR2, Electron Microscopy Sciences) holey carbon grids, respectively. Grids were blotted for 2 s at 100% humidity and flash frozen in liquid ethane. Grids were loaded onto a Glacios (FEI) electron microscope operating at 200 kV equipped with a Falcon IV detector (Thermo Fisher Scientific). Movies were recorded using the EPU software (Thermo Fisher Scientific) with a pixel size of 0.94 Å and a nominal defocus value between −0.6 to −2.4 µm. Data were collected with a dose of ~4.72 electrons per Å$^2$ per second, and each movie was recorded by 48 frames (203 ms per frame) for a 9.77 s exposure.

For the WT *Ec*MscK, a total of 3 ul of purified channel protein (6.5 mg ml$^{-1}$) was applied onto glow-discharged copper Quantifoil R2/2 holey carbon grids. Grids were loaded onto a Krios (FEI) electron microscope operating at 300 kV equipped with a Gatan K2 detector. Movies were recorded using the EPU software with a pixel size of 1.1 Å and a nominal defocus value between −1.0 to −2.5 µm. Data were collected with a dose of ~8.26 electrons per Å$^2$ per second, and each movie was recorded by 40 frames (200 ms per frame) for an 8 s exposure.

## Image processing and map calculation

Images stacks were first aligned and dose-weighted using patch-based motion correction, and then subjected to patch-based contrast transfer function (CTF) determination in CryoSPARC 3.3.1[37]. Following motion correction and CTF estimation, low-quality images were manually removed from the datasets on the basis of estimated CTF resolution and cross-correlation. Particles were picked using a blob-based autopicker and subjected to 2D classification. 2D classes corresponding to MscK were combined and used to generate an ab initio model with C1 symmetry, which was refined with C7 symmetry, lowpass filtered to 15 Å, and then used as a template for subsequent heterogeneous refinement. Classes from heterogeneous refinement were subjected to iterative nonuniform refinement[54], CTF refinement, and per-particle local motion correction before sharpening with DeepEMhancer[55], when map quality had been optimized. For each reconstruction, a focused refinement of the channel core, including TM9-11 and the CTD, was performed to obtain higher-resolution details of the channel conduction pathway.

For the *Ec*MscK G924S mutant, 2823 movies collected in 150 mM NaCl and 4010 movies in 150 mM KCl were combined and subjected to patch-based motion correction and CTF estimation in cryoSPARC[37]. The exposures were manually curated to remove micrographs with poor quality, resulting in 2481 micrographs from the NaCl dataset and 3874 from the KCl dataset. Blob-based autopicking was performed on the curated micrographs, with particle diameters between 200 and 300 Å, resulting in 1,138,139 particles. These particles were subjected to 2D classification, and the classes from 319,952 particles corresponding to intact MscK channels were used as templates for further template-based particle picking. Particles selected from template-based picking were subjected to 2D classification, resulting in 482,410 particles corresponding to intact MscK channels. About 25,072 duplicate particles were removed from this selection (distance within 140 Å). The remaining 457,338 particles were used to generate an initial model, which was subsequently refined with C7 symmetry and lowpass filtered to 15 Å. This density map was used to conduct heterogeneous refinement with six classes, which included open, closed, and intermediate conformations of the channel and junk particles. Each class was subjected to iterative rounds of nonuniform refinement with C7 symmetry followed by per-particle local motion correction and CTF refinement until no further improvement of the map quality. The half maps from the final round of nonuniform refinement were sharpened using deepEMhancer on the highRes preset[55]. Local resolution estimation was performed on the final round of nonuniform refinement.

For WT *Ec*MscK, 2188 micrographs were subjected to patch-based motion correction and CTF estimation. Manual curation revealed that the dataset was of exceptional quality and had only minimal ice contamination−2120 of 2188 micrographs were utilized. Blob-based autopicking was performed with particle diameters between 200 and 300 Å, resulting in 435,710 selected particles. Particles were subjected to 2D classification and 167,237 particles corresponding to templates for intact *Ec*MscK channels were used for template-based autopicking. After 2D classification, 218,385 particles were identified that corresponded to intact channels and were used to generate an ab initio model with C1 symmetry, which was subsequently refined with C7 symmetry and lowpass filtered to 15 Å. Subsequent heterogeneous refinement with four classes resulted in two classes representing the closed conformation and the other two classes representing channels with weak and broken density. The two classes representing the closed conformation were combined, subjected to nonuniform refinement, and followed by per-particle local motion correction and nonuniform refinement. CTF refinement did not improve map quality, so the half maps from the original nonuniform refinement were sharpened with deepEMhancer on the highRes preset to generate the final map.

The AlphaFold model of *Ec*MscK[56] was used as an initial model guiding atomic model building and was manually segmented to fit into the density map of the higher-resolution G924S closed state. After the deletion of portions that were not present in the density, the model was further trimmed with the deletion of side chains that were not well resolved. Segments of the map that could not be registered with certainty were built as polyalanine models. Atomic models were manually adjusted in COOT[57] before real-space refinement in PHENIX[58], and the final quality was assessed using MolProbity[59]. The structural illustration was prepared using UCSF ChimeraX[60]. Figure 6 was created with ChimeraX and BioRender.com.

## Giant spheroplasts preparation

*E. coli* giant spheroplasts were prepared from the pB113 cell line (*mscK-*, *mscS-*)[32], using each of the constructs: pET300-*Ec*MscK (WT), pET300-*Ec*MscK W914A, or pET300-*Ec*MscK G924S following the published protocol[61]. Briefly, the cells were grown in Luria-Bertani (LB) broth containing carbenicillin and cephalexin to form filamentous cells for about 1.5 h. Cells were induced by 1 mM Isopropyl β-D-1-thiogalacto-pyranoside (IPTG, I2481C100, GoldBio) for 1 h and harvested by

centrifugation. The pellets were gently resuspended in 1.25 ml of 1 M sucrose, to which 75 μl of 1 M Tris pH 8.0, 50 μl of 5 mg ml⁻¹ lysozyme (L-040-25, GoldBio), 15 μl of 5 mg ml⁻¹ DNase, and 50 μl of 125 mM EDTA pH 7.8 were added. Lysozyme digestion was carried out for 12 to 15 min and was terminated by the addition of 0.5 ml of stop solution (875 μl of 1 M sucrose, 125 μl of water, 20 μl of 1 M MgCl₂, 10 μl of 1 M Tris pH 8.0). The reactions were added into the tubes with 7 ml of ice-cold dilution solution (10 mM MgCl₂, 10 mM Tris pH 8.0 in 1 M sucrose). Spheroplasts were pelleted by centrifugation at 4 °C, gently resuspended in ~0.5 ml of dilution solution, aliquoted, and stored at −20 °C.

### Electrophysiology

For the patch-clamp experiments, symmetric KCl or NaCl buffers were used (200 mM KCl or NaCl, 90 mM MgCl₂, 2 mM CaCl₂, 5 mM HEPES, pH 7.2) with bath solution supplemented with 400 mM sucrose. Upon patch excision from the spheroplast membrane (inside-out configuration), suction was manually applied to the pipette via syringe in order to generate lateral tension in a membrane. Recordings were made and digitized with the Axopatch 1D patch-clamp amplifier, the Digidata 1320 digitizer (Molecular Devices), and the PM-015R pressure monitor (World Precision Instruments). Data were collected at 1 kHz, lowpass filtered at 200 or 500 Hz, and analyzed with the pClamp software suite (Molecular Devices). The pipettes with ~1–2 MΩ resistance were fabricated from the Kimble Chase soda lime glass with a Sutter P-96 puller (Sutter Instruments). All measurements were carried out at −30 mV membrane potential or as specified in the text.

Tension sensitivity of the WT MscK and mutants was assayed using the ratio of $P_{McsL}/P_{MscK}$, which was calculated as the ratio of pressures at which the first MscL and MscK channels were activated, respectively, in response to a slow ~4 mmHg s⁻¹ pressure ramp. Higher $P_{McsL}/P_{MscK}$ ratio indicated higher tension sensitivity of a given $Ec$MscK construct with respect to endogenous $Ec$MscL, which was used as an internal calibration.

### Reporting summary

Further information on research design is available in the Nature Portfolio Reporting Summary linked to this article.

## Data availability

All data needed to evaluate the conclusions in the paper are presented in the paper or the Supplementary Materials. The cryo-EM maps have been deposited to Electron Microscopy Data Bank (EMDB) with accession codes EMD-26851 (WT closed), EMD-26823 (G924S closed), EMD-26845 (G924S open), EMD-26854 (G924S intermediate), EMD-26872 (WT, focused refinement), EMD-26875 (G924S closed, focused refinement), EMD-26877 (G924S open, focused refinement), and EMD-26876 (G924S intermediate, focused refinement). Atomic coordinates have been deposited to the Protein Data Bank (PDB) with accession codes 7UW5 (G924S closed) and 7UX1 (G924S open). The source data underlying Supplementary Fig. 2 are provided as a Source Data file. Source data are provided with this paper.

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

## Acknowledgements

We thank members of the Yuan lab for the discussion. This work was supported by NIH grants R01NS099341 and R01GM143440 (to P.Y.).

## Author contributions

J.M. performed biochemical preparations. J.M., B.T.S., and J.A.J.F. collected cryo-EM data. J.M. analyzed cryo-EM data. G.M conducted electrophysiology experiments. P.Y. conceived and supervised the work. J.M., G.M., and P.Y. prepared the manuscript. All authors edited the manuscript.

## Competing interests

The authors declare no competing interests.
