## [Peer Review File · Nature Communications]

Structural basis for mechanotransduction in a potassium-dependent mechanosensitive ion channelEditorial Note: Parts of this Peer Review File have been redacted as indicated to remove third party material where no permission to publish were obtained

Reviewers' Comments:

Reviewer #1:

Remarks to the Author:

In this manuscript, Mount et al. present structures of a K⁺ dependent bacterial mechanosensitive channel MscK in open, closed, and intermediate channel states. This is an important paper because it furthers our understanding of mechanical gating in the diverse MscS-like family and of mechanosensitive channels generally. The structures are very interesting and this is, overall, a nicely executed and presented study. I only recommend that the following issues be addressed prior to publication.

1. Could the authors expand on their discussion of energetics involved in channel gating in the discussion? This would add considerable interest for the field. With the structures you have determined here, you could for instance estimate the energy difference between states due to (1) area expansion and (2) midplane bending (see e.g. Ursell, T. et al, Role of Lipid Bilayer Mechanics in Mechanosensation. p37–70 in Mechanosensitive Ion Channels, 2008) and see whether these two factors explain the sensitivity of MscK compared to e.g. MscS/L.
2. Please clarify which maps were used for real-space refinement. If Deep EMhancer maps were used for refinement (in addition to guiding model building), final refinement should be rerun using traditionally sharpened maps to avoid bias.
3. The statement “Consistent with functional characterization, both NaCl and KCl conditions yielded apparently distinct conformations” (p6) does not make sense to me and probably should be changed. Functional data shows the channel is closed at low tension (basal patch tension is presumably on the order of a few mN/m) so the simple expectation is that you would observe closed structures. Related to this, it is interesting that similar proportions of closed and open states are observed in KCl and NaCl since the channel seems to open more readily in KCl at similar pressures. Can the authors speculate on why they see different conformations in their datasets and why similar distributions are observed?
4. “Gating tension threshold” (page 5 and Extended Data 2b) should be replaced with “mechanical activation threshold” or “pressure activation threshold” since pressure was the stimulus measured and corresponding tension was not measured/calculated (and is not necessarily linearly related to pressure in these experiments).
5. In Fig 3d, is the pore diameter between 100-120Å irrelevant because it's below the side portals in the cytoplasmic domain that are wide enough for ion passage in both conformations? Perhaps the position of these could be indicated on the plot and their diameter included in Extended Data Fig. 7 for reference?
6. Related to Extended Data Fig. 7, an overlay of the core regions of MscK, MscS, and MSL1 and rmsds for C α deviations would be helpful for understanding how similar these channels are.
7. Can the authors offer additional explanation or speculation as why the G924S mutant results in gain of function? Perhaps accompanied with a zoomed-in view to show rearrangements of gating residues described on page 9?
8. Can the authors speculate as to which residues or interactions are particularly important for K⁺ sensing by MscK? Perhaps guided by the AlphaFold model since the periplasmic region was not modeled? Are there other gain of function mutants known that provide any insight given the new structural insights here?

Reviewer #2:

Remarks to the Author:

Mount et al present the open and closed CryoEM structure of MscK. The authors trap MscK in the open state using known gain of function point mutations that increase open probability in the absence of potassium. Importantly, the authors use their structure to design a point mutation (W914) that decreases the pressure threshold for MscK and propose a model for MscK channel gating by mechanical force. The structural changes are dramatic and clear. Trapping open and closed states is highly significant for a mechanosensitive channel. The paper can be recommended for publication in Nature Communications after addressing some minor issues with the model and some additional clarification.

Major points

Point 1: Extended data Fig. 2 needs calculations for statistical significance in both a and b. If in b, the W914A is not statistically different than WT, this needs to be stated in the main text. Also, in b, the protocol of how PmscL was determined for calculating the PMscL/PMscK ratio seems to be missing. Please add a description of what was actually done. And please also comment if when MscK was in NaCl was MscL also in NaCl. Does NaCl vs KCl affect MscL gating tension thresholds?

Point 2: The authors conclude/speculate that tension in the membrane causes a "flattening" of the protein and that the loss of curvature provides the energy for opening. This is indeed plausible based on known properties of midplane bending and the authors are welcome to speculate based on their excellent structural data. However, hydrophobic mismatch is also a possibility and wasn't considered. In Figure 3b the authors measured a 19Å decrease in the thickness of the membrane spanning region of MscK in the open state. Based on this structural change, hydrophobic mismatch likely plays a role in gating the channel.

The authors need to consider the effect on hydrophobic mismatch in their mechanism and either include it or justify its exclusion in favor of midplane bending. The conceptual analysis for these two mechanisms was presented for MscL by Ursell et. al. 2008[1] and should apply to MscK. A figure comparing the mathematical difference from midplane bending (protein curvature) vs hydrophobic mismatch is shown in Figure 7b and 7c-d respectively of Ursell et. al and included here as an attachment. The effect of tension on hydrophobic mismatch is shown in Figure 7e-f of Ursell. The authors need to estimate the hydrophobic thickness of MscK relative to the average bacterial membrane. If in the closed state, MscK is thicker than the average bacterial membrane, then hydrophobic mismatch shown in Ursell Figure 7f applies and it should be discussed and included as a possible mechanism for MscK opening.

Minor points

Point 3. The authors depict the membrane in Fig. 6a pointed up, yet the slope of the protein is pointed down with a 30 degree slope. The lipids in Fig. 6a should follow the slope of the protein surface and point downward as shown by Ursell et. al. in Figure 7b.

Reference

1 Ursell, T. et al. (2008) The Role of Lipid Bilayer Mechanics in Mechanosensation. *Mechanosensitive Ion Channels* 1, 37–80

Reviewer #3:

Remarks to the Author:

The Msc family of prokaryotic proteins have been the long-studied archetypal proteins for understanding the molecular mechanisms of mechanosensitive channels. Here, Mount et al use electrophysiology and cryo-EM to study a lesser known family member, MscK, whose activation by

potassium ions and unique structural features (additional TM helices) make it an interesting target for study.

The strengths of this paper lies in the simplicity of the data, which is very beautiful. Using a GOF mutation the authors are able to obtain multiple conformations of the channel. MscK is clearly curved in the closed conformation and flattens and expands upon opening. This adds to the growing data in the field about mechanisms by which ion channels can sense membrane tension and bolsters current thinking about non-planarity of transmembrane helices (and the resulting local membrane deformation) as a means to increase tension sensitivity.

My concerns with the paper are mostly minor and concern the text which in some parts lacks nuance and thoughtfulness. I think these points can be remedied very easily and will strengthen the paper as a key contribution to the field of understanding molecular mechanosensation, which it is.

1. Although the data speaks for itself (the authors clearly obtain an open conformation within their cryo-EM dataset) it is somewhat puzzling to me that the authors find that around 30% of their particles are in the open conformation. This suggests a high open probability in detergent micelles, which of course are tensionless. In their spheroblast electrophysiology recordings the G924S mutant appears to have an open probability of 0 at 0mmHG (which will still be at a significant membrane tension, as has been measured by others). Maybe if ephys recordings are done for longer they would see spontaneous openings of the channel but otherwise I wonder if the authors can comment on what might be happening here - why they can obtain an open conformation in a tensionless system?

2. Similar to this line of thinking by ephys recordings the authors find that the lower gating threshold for G924S occurs only in KCl conditions but state that their cryo-EM datasets, including relative particle distributions, for NaCl and KCl are identical. It seems that detergent vs lipid has very significant effects on the gating properties and I think it's important for the authors to at least mention this (it may be related to lipid interactions with the protein in the large unoccupied spaces between protein subunits they observe in their structure).

3. Although only speculation I would be more careful about any correlation of the two gates pertaining to distinct gating mechanisms. These gates are not "distant" (they are 7 residues apart), and they are part of the same helix that seems to rotate as one during the opening conformational change observed by the authors.

4. I do not agree, at least from what I see, that W914A destabilizes both the closed and open conformations. It is hard to say from a few representative ephys traces but it appears to me that the closed dwell time is similar for W914A and the WT channel and only the open dwell time is markedly decreased. Proper single-channel analysis, which should be simple for these types of recordings, would settle this, otherwise I would be more careful with phrasing. On this point it would be worth commenting more on the interactions of W914 and the other hydrophobic gating residues in the open state. Presumably in the closed state they shield one another from water, forming a closed gate, and in the open state are stabilized by different hydrophobic interactions. In this way one could imagine how the W914A mutation would lead to loss of an interaction in the open state, destabilizing this conformation and reducing open dwell time as appears in the ephys traces. I would include some depiction of these interactions in the extended data figures - perhaps in place of extended data figure 8c which to me is convoluted and hard to interpret.

5. Although I appreciate the inclusion of all conformations obtained by their cryo-EM analysis it is unclear to me what an "intermediate" conformation of the channel is. Although limited by resolution the pore-lining helices appear further from one another than in the closed conformation, do the authors expect this pore to be conductive? Are there sub-conductance states for MscK?

6. It is clear that MscK gates at lower membrane tensions than MscL in spheroblast ephys recordings.

It is tempting to speculate that this is due, at least partly, to the additional non-planar helices and subsequent membrane deformation of the channel. The notion of such a membrane "footprint" and its contribution to tension sensitivity has been well worked out (see Haselwandter and Mackinnon, <https://doi.org/10.7554/eLife.41968>). I think the discussion would benefit from some elaboration on this point. To me at least it is remarkable to see how nature utilizes local deformation of the bilayer to amplify tension sensitivity in different ion channels and is one of the standout points of the study.

Reviewer #4:

Remarks to the Author:

In this work, Mount et al determined the cryo-EM structures of a bacterial mechanosensitive channel, MscK, in closed, open, and intermediate states. Structural comparison clearly showed the remarkable conformational changes between closed and open states, including the increase of the heptamer diameter and the decrease of the mid-plane bending angle. Based on the structural observations, they proposed a mechanotransduction mechanism for the mechanosensitive channels. The authors captured three distinct conformations of MscK that strongly support their conclusions. There is no concern or question for the structural part. Several minor suggestions from this referee may help to improve the manuscript:

1. In Figure 2, an additional panel showing how twisted TM11 forms the pore may benefit the readers who don't work in the ion channel field.
2. In Figure 3c and e, please use a bright color to show W914, V921 and F925.
3. In Extended Data Table 1, please edit the "PDB xxxx".

REVIEWER COMMENTS

Reviewer #1 (Remarks to the Author):

In this manuscript, Mount et al. present structures of a K⁺ dependent bacterial mechanosensitive channel MscK in open, closed, and intermediate channel states. This is an important paper because it furthers our understanding of mechanical gating in the diverse MscS-like family and of mechanosensitive channels generally. The structures are very interesting and this is, overall, a nicely executed and presented study. I only recommend that the following issues be addressed prior to publication.

Point 1: Could the authors expand on their discussion of energetics involved in channel gating in the discussion? This would add considerable interest for the field. With the structures you have determined here, you could for instance estimate the energy difference between states due to (1) area expansion and (2) midplane bending (see e.g. Ursell, T. et al, Role of Lipid Bilayer Mechanics in Mechanosensation. p37–70 in Mechanosensitive Ion Channels, 2008) and see whether these two factors explain the sensitivity of MscK compared to e.g. MscS/L.

Response: We appreciate this point and have now expanded our discussion of energetics due to (1) area expansion and (2) midplane bending based on the framework laid out by Ursell et al, Role of Lipid Bilayer Mechanics in Mechanosensation. p37–70 in Mechanosensitive Ion Channels, 2008). These two factors, without consideration of intrinsic free energy difference between the two states of the channel protein, give rise to a midpoint membrane tension of 0.16 k_BT/nm², which is much lower than the reported midpoint tension of MscL (2.5 k_BT/nm²). The estimation implies that MscK could display much higher tension sensitivity than MscL. However, contribution of the intrinsic free energy difference between the closed and open conformations is not accounted for in this estimation. Notably, the closed and open MscK structures suggest that additional free energy cost must be paid to expand the periplasmic gating ring, thus a higher midpoint membrane tension would be expected. We have now included the following discussion on page 13.

“The difference of free energy of midplane bending between the closed (in-plane radius R_c , midplane bending angle θ_c) and open (in-plane radius R_o , midplane bending angle θ_o) states can be approximated by

$$\Delta G_{bending} = \pi(\theta_c^2 R_c - \theta_o^2 R_o) \sqrt{\sigma K_b} \quad (2),$$

where K_b is the bilayer bending modulus $\sim 20 k_B T$ (k_B is Boltzmann’s constant, and T is the temperature)⁵⁰. The contribution of membrane deformation to tension sensitivity can be estimated by the midpoint membrane tension ($\sigma_{1/2}$), at which the open probability of the channel is 0.5. Without consideration of $\Delta G_{\sigma=0}$, $\sigma_{1/2}$ is determined by

$$\pi(\theta_c^2 R_c - \theta_o^2 R_o) \sqrt{\sigma_{1/2} K_b} - \sigma_{1/2} \Delta A = 0 \quad (3).$$

Thus, the midpoint tension is calculated by

$$\sigma_{1/2} = \frac{(\theta_c^2 R_c - \theta_o^2 R_o)^2 K_b}{4 R_c^2 \Delta R^2} \quad (4).$$

In MscK, the parameters deduced from the closed and open structures result in an estimated midpoint membrane tension of 0.16 k_BT/nm², much lower than the reported midpoint tension of MscL (2.5 k_BT/nm²)⁵¹, which is primarily determined by energetics of hydrophobic mismatch, rather than midplane bending, between the TM helices and the surrounding lipid bilayer. Opening of MscK is essentially achieved by rigid body movement of the TMD, which

appears to maintain a constant membrane thickness matching that of the bacterial inner membrane. Thus, hydrophobic mismatch may not contribute substantially to tension sensitivity of MscK. This analysis implies that MscK could display much higher tension sensitivity than MscL. Analogously, in Piezo channels, the extendedly curved TMD induces a curved 'membrane footprint' that amplifies tension sensitivity⁵². In our analysis, contribution of the intrinsic free energy difference between the closed and open conformations, $\Delta G_{\sigma=0}$, is not included. Notably, the closed and open MscK structures suggest that additional free energy cost must be paid to expand the periplasmic gating ring, thus a higher midpoint membrane tension would be expected."

Point 2: Please clarify which maps were used for real-space refinement. If Deep EMhancer maps were used for refinement (in addition to guiding model building), final refinement should be rerun using traditionally sharpened maps to avoid bias.

Response: The DeepEMhancer maps were used for refinement. We have now refined the final models using traditionally sharpened maps and have updated the coordinates deposited to the databank.

Point 3: The statement "Consistent with functional characterization, both NaCl and KCl conditions yielded apparently distinct conformations" (p6) does not make sense to me and probably should be changed. Functional data shows the channel is closed at low tension (basal patch tension is presumably on the order of a few mN/m) so the simple expectation is that you would observe closed structures. Related to this, it is interesting that similar proportions of closed and open states are observed in KCl and NaCl since the channel seems to open more readily in KCl at similar pressures. Can the authors speculate on why they see different conformations in their datasets and why similar distributions are observed?

Response: The reviewer is correct, and we appreciate this point. We have removed "Consistent with functional characterization" and now state the following:

"Notably, both NaCl and KCl conditions yielded apparently distinct conformations that presumably represent the closed and open states..."

The G924S channel is closed at low tension in excised patches. However, for single-particle cryo-EM studies, the channels were extracted from cell membranes and solubilized in detergents, which are completely different from biological membranes. Thus, the distribution of functional states of the purified channel in detergents could deviate from that observed in excised membrane patches. We can compare channel activity of the wild type and G924S in excised patches and compare conformations of the wild type and G924S in detergents in single particle analysis. Compared with the wild-type channel, the increased activity for G924S in membrane patches is consistent with observation of open conformation in single particle cryoEM studies.

For instance, in our previous studies of the MSL1 channel (a plant homolog of MscS), the gain-of-function mutant, A320V, also requires tension to gate. However, the single particles in cryo-EM samples all represent the open conformation. Clearly, we acknowledge that there is discrepancy between electrophysiology in membrane patches and single particle cryoEM experiments with purified channels in detergent or lipid environments.

Reference:

1. Deng Z, Maksaev G, Schlegel AM, Zhang J, Rau M, Fitzpatrick JAJ, Haswell ES, Yuan P. Structural mechanism for gating of a eukaryotic mechanosensitive channel of small conductance. Nat Commun. 2020 Jul 23;11(1):3690. doi: 10.1038/s41467-020-17538-1.

To make this point clear, we have now also included the following discussion on page 14.

“Distinct conformations representing the closed, intermediate, and open states and a similar distribution of these conformational states were obtained in both the Na⁺ and K⁺ conditions in cryo-EM experiments. This contrasts with channel activities observed in excised membrane patches, in which the mutant channel, G924S, remained closed in the absence of applied pressure to the excised patches and displayed a lower gating pressure threshold in K⁺, than Na⁺, recording conditions. These discrepancies likely originate from detergent micelles, which are necessary to isolate channel protein for cryo-EM but drastically differ from the excised cell membranes in electrophysiology experiments. Notably, the detergent environment, devoid of lipids, appears to additionally modify the gating properties and does not necessarily recapitulate channel properties in the lipid bilayer, as has been observed in MscS²⁴. In this work, detergent-solubilized MscK channels, in the absence of tension, have provided unanticipated insights into channel gating and allowed us to infer the mechanotransduction process in an ion channel.”

Point 4: “Gating tension threshold” (page 5 and Extended Data 2b) should be replaced with “mechanical activation threshold” or “pressure activation threshold” since pressure was the stimulus measured and corresponding tension was not measured/calculated (and is not necessarily linearly related to pressure in these experiments).

Response: We thank the reviewer for this point and have now changed “gating tension threshold” to “pressure activation threshold”.

Point 5: In Fig 3d, is the pore diameter between 100-120Å irrelevant because it’s below the side portals in the cytoplasmic domain that are wide enough for ion passage in both conformations? Perhaps the position of these could be indicated on the plot and their diameter included in Extended Data Fig. 7 for reference?

Response: We agree with the reviewer. The pore diameter between 100-120 Å is irrelevant because it is below the side portals that are wide enough for ion passage in both conformations. To make this point clear, we have now indicated the position of these side portals on the plot in the new Fig. 3d and indicated the narrowest dimension of the side portal in the new Extended Data Fig. 7 for reference.

new Fig. 3c-e

new Extended Data Fig. 7b

Point 6: Related to Extended Data Fig. 7, an overlay of the core regions of MscK, MscS, and MSL1 and rmsds for C α deviations would be helpful for understanding how similar these channels are.

Response: This is a great point. We have now also included overlays of MscK/MscS; MscK/MSL1 by the conserved cytoplasmic CTD and indicated the rmsds for Ca deviations for the CTD in the new Extended Data Figs. 7f and 7g. We have now also included the following on page 7.

“Notably, while the conserved CTDs align well with each other, the TM helices are distinctly arranged in the EcMscK channel (Extended Data Fig. 7 f, g). In comparison with EcMscS and AtMSL1, the TMD in EcMscK rotates counterclockwise viewed from the periplasmic or extracellular side, which pivots at the helical kink.”

Point 7: Can the authors offer additional explanation or speculation as why the G924S mutant results in gain of function? Perhaps accompanied with a zoomed-in view to show rearrangements of gating residues described on page 9?

Response: In our last submission, we have briefly speculated why the G924S mutant results in gain of function on page 13 in Discussion (now second paragraph on page 15). It appears that the pore-lining helix in various MscS homologs is enriched in small-sized glycine and alanine residues, which enable tight helix-helix associations critical for stabilization of the closed conformation. Consequently, introduction of a bulkier side chain at one of these locations can promote channel opening, likely through destabilization of the closed conformation. We have previously tested this hypothesis in our *AfMSL1* studies by introduction of the A320V mutant, which also resulted in an open conformation (Reference: Deng et al, Nature Communications 2020). To make this point clear, we have now also included a zoomed-in view to show the tight packing interactions in the pore helix of MscK in the new Extended Data Fig. 9.

Extended Data Fig. 9 | G924S and helix packing. a,b, Packing of pore-lining helices in the closed (a) and open (b) states. Glycine and alanine residues in the pore-lining helices are highlighted as spheres. Positions of the GOF mutations (G922 and G924) are labeled.

Reference:

Deng Z, Maksaev G, Schlegel AM, Zhang J, Rau M, Fitzpatrick JAJ, Haswell ES, Yuan P. Structural mechanism for gating of a eukaryotic mechanosensitive channel of small conductance. Nat Commun. 2020 Jul 23;11(1):3690. doi: 10.1038/s41467-020-17538-1.

Point 8: Can the authors speculate as to which residues or interactions are particularly important for K⁺ sensing by MscK? Perhaps guided by the AlphaFold model since the periplasmic region was not modeled? Are there other gain of function mutants known that provide any insight given the new structural insights here?

Response: This is an excellent point and an important future question to address. To understand the K⁺ sensing mechanism, we have been trying X-ray crystallography with the purified periplasmic domain, which is ongoing and may require tremendous effort. We have attempted to identify potential K⁺ regulation sites using the AlphaFold model in the periplasmic region, but this has been very challenging. Unfortunately, MscK has not been heavily studied, and information of known functional mutants is very limited. Therefore, we feel that this is an important future direction but is out of the scope of our current study.

Reviewer #2 (Remarks to the Author):

Mount et al present the open and closed CryoEM structure of MscK. The authors trap MscK in the open state using known gain of function point mutations that increase open probability in the absence of potassium. Importantly, the authors use their structure to design a point mutation (W914) that decreases the pressure threshold for MscK and propose a model for MscK channel gating by mechanical force. The structural changes are dramatic and clear. Trapping open and closed states is highly significant for a mechanosensitive channel. The paper can be recommended for publication in Nature Communications after addressing some minor issues with the model and some additional clarification.

Major points

Point 1: Extended data Fig. 2 needs calculations for statistical significance in both a and b. If in b, the W914A is not statistically different than WT, this needs to be stated in the main text. Also, in b, the protocol of how P_{mScL} was determined for calculating the P_{mScL}/P_{mScK} ratio seems to be missing. Please add a description of what was actually done. And please also comment if when MscK was in NaCl was MscL also in NaCl. Does NaCl vs KCl affect MscL gating tension thresholds?

Response: We appreciate these points. We have now included calculations for statistical significance (Student's t-test) in both **a** and **b** in the new Extended Data Fig. 2. The protocol for calculating the P_{mScL}/P_{mScK} ratio has been described in detail in **Methods**.

"Tension sensitivity of the WT MscK and mutants was assayed using the ratio of P_{mScL}/P_{mScK}, which was calculated as the ratio of pressures at which the first MscL and MscK channels were activated, respectively, in response to a slow ~4 mmHg s⁻¹ pressure ramp."

As both MscK and endogenous MscL channels were always present in the same membrane patches, these channels were subjected to the same recording conditions. MscL gating tension thresholds are the same in NaCl or KCl conditions (Li & Blount 2002; Booth & Blount 2012). To make it clear, we have now included the following on page 5:

*“Endogenous *E. coli* MscL (EcMscL) channels, which open at essentially the same high tension threshold in Na⁺ or K⁺ conditions^{19,32}, provide calibration of tension in excised membrane patches.”*

new Extended Data Fig. 2

Point 2: The authors conclude/speculate that tension in the membrane causes a “flattening” of the protein and that the loss of curvature provides the energy for opening. This is indeed plausible based on known properties of midplane bending and the authors are welcome to speculate based on their excellent structural data. However, hydrophobic mismatch is also a possibility and wasn’t considered. In Figure 3b the authors measured a 19 Å decrease in the thickness of the membrane spanning region of MscK in the open state. Based on this structural change, hydrophobic mismatch likely plays a role in gating the channel.

The authors need to consider the effect on hydrophobic mismatch in their mechanism and either include it or justify its exclusion in favor of midplane bending. The conceptual analysis for these two mechanisms was presented for MscL by Ursell et. al. 2008[1] and should apply to MscK. A figure comparing the mathematical difference from midplane bending (protein curvature) vs hydrophobic mismatch is shown in Figure 7b and 7c-d respectively of Ursell et. al and included here as an attachment. The effect of tension on hydrophobic mismatch is shown in Figure 7e-f of Ursell. The authors need to estimate the hydrophobic thickness of MscK relative to the average bacterial membrane. If in the closed state, MscK is thicker than the average bacterial membrane, then hydrophobic mismatch shown in Ursell Figure 7f applies and it should be discussed and included as a possible mechanism for MscK opening.

Response: We appreciate this excellent point. However, membrane deformation induced by MscK mainly comes from midplane bending, but not the thickness, of the surrounding bilayer. Thus, we do not think that hydrophobic mismatch significantly contributes to the energetics in MscK gating. The 19 Å decrease refers to the overall dimension of the transmembrane domain, not the membrane thickness surrounding the transmembrane segments (i.e., transmembrane

helices). This 19 Å decrease is a consequence of the ‘flattening’ of the curved transmembrane domain. The lengths of the transmembrane helices are close to an average membrane thickness of ~37Å. Therefore, we think that the membrane thickness around the transmembrane domain of MscK is similar in the closed and open states. As also suggested by Reviewer 1, we have now included more detailed discussion about the energetics of midplane bending and have included our hypothesis that hydrophobic mismatch may not play an important role in MscK gating.

“The difference of free energy of midplane bending between the closed (in-plane radius R_c , midplane bending angle θ_c) and open (in-plane radius R_o , midplane bending angle θ_o) states can be approximated by

$$\Delta G_{bending} = \pi(\theta_c^2 R_c - \theta_o^2 R_o) \sqrt{\sigma K_b} \quad (2),$$

where K_b is the bilayer bending modulus $\sim 20 k_B T$ (k_B is Boltzmann’s constant, and T is the temperature)⁵⁰. The contribution of membrane deformation to tension sensitivity can be estimated by the midpoint membrane tension ($\sigma_{1/2}$), at which the open probability of the channel is 0.5. Without consideration of $\Delta G_{\sigma=0}$, $\sigma_{1/2}$ is determined by

$$\pi(\theta_c^2 R_c - \theta_o^2 R_o) \sqrt{\sigma_{1/2} K_b} - \sigma_{1/2} \Delta A = 0 \quad (3).$$

Thus, the midpoint tension is calculated by

$$\sigma_{1/2} = \frac{(\theta_c^2 R_c - \theta_o^2 R_o)^2 K_b}{4R_c^2 \Delta R^2} \quad (4).$$

In MscK, the parameters deduced from the closed and open structures result in an estimated midpoint membrane tension of 0.16 $k_B T/nm^2$, much lower than the reported midpoint tension of MscL (2.5 $k_B T/nm^2$)⁵¹, which is primarily determined by energetics of hydrophobic mismatch, rather than midplane bending, between the TM helices and the surrounding lipid bilayer. Opening of MscK is essentially achieved by rigid body movement of the TMD, which appears to maintain a constant membrane thickness matching that of the bacterial inner membrane. Thus, hydrophobic mismatch may not contribute substantially to tension sensitivity of MscK. This analysis implies that MscK could display much higher tension sensitivity than MscL. Analogously, in Piezo channels, the extendedly curved TMD induces a curved ‘membrane footprint’ that amplifies tension sensitivity⁵². In our analysis, contribution of the intrinsic free energy difference between the closed and open conformations, $\Delta G_{\sigma=0}$, is not included. Notably, the closed and open MscK structures suggest that additional free energy cost must be paid to expand the periplasmic gating ring, thus a higher midpoint membrane tension would be expected.”

Minor points

Point 3: The authors depict the membrane in Fig. 6a pointed up, yet the slope of the protein is pointed down with a 30 degree slope. The lipids in Fig. 6a should follow the slope of the protein surface and point downward as shown by Ursell et. al. in Figure 7b.

Response: We appreciate this point and have now modified Fig. 6a to better describe the lipids curving downward.

new Fig. 6

Reviewer #3 (Remarks to the Author):

The Msc family of prokaryotic proteins have been the long-studied archetypal proteins for understanding the molecular mechanisms of mechanosensitive channels. Here, Mount et al use electrophysiology and cryo-EM to study a lesser known family member, MscK, whose activation by potassium ions and unique structural features (additional TM helices) make it an interesting target for study.

The strengths of this paper lies in the simplicity of the data, which is very beautiful. Using a GOF mutation the authors are able to obtain multiple conformations of the channel. MscK is clearly curved in the closed conformation and flattens and expands upon opening. This adds to the growing data in the field about mechanisms by which ion channels can sense membrane tension and bolsters current thinking about non-planarity of transmembrane helices (and the resulting local membrane deformation) as a means to increase tension sensitivity.

My concerns with the paper are mostly minor and concern the text which in some parts lacks nuance and thoughtfulness. I think these points can be remedied very easily and will strengthen the paper as a key contribution to the field of understanding molecular mechanosensation, which it is.

Point 1: Although the data speaks for itself (the authors clearly obtain an open conformation within their cryo-EM dataset) it is somewhat puzzling to me that the authors find that around 30% of their particles are in the open conformation. This suggests a high open probability in detergent micelles, which of course are tensionless. In their spheroblast electrophysiology recordings the G924S mutant appears to have an open probability of 0 at 0mmHG (which will still be at a significant membrane tension, as has been measured by others). Maybe if ephys recordings are done for longer they would see spontaneous openings of the channel but otherwise I wonder if the authors can comment on what might be happening here - why they can obtain an open conformation in a tensionless system?

Response: We appreciate this excellent point, which has also been noted by **Reviewer 1 (Point 3)**. The G924S channel is closed at low tension in excised patches. However, for single-particle cryo-EM studies, the channels were extracted from cell membranes and solubilized in detergents, which are completely different from biological membranes. Thus, the distribution of functional states of the purified channel in detergents could deviate from that observed in excised membrane patches.

For instance, in our previous studies of the MSL1 channel (a plant homolog of MscS), the gain-of-function mutant, A320V, also requires tension to gate. However, the single particles in cryo-EM samples all represent the open conformation. Clearly, we acknowledge that there is discrepancy between electrophysiology in membrane patches and single particle cryoEM experiments with purified channels in detergent or lipid environments.

Reference:

1. Deng Z, Maksaev G, Schlegel AM, Zhang J, Rau M, Fitzpatrick JAJ, Haswell ES, Yuan P. Structural mechanism for gating of a eukaryotic mechanosensitive channel of small conductance. Nat Commun. 2020 Jul 23;11(1):3690. doi: 10.1038/s41467-020-17538-1.

To make this point clear, we have now also included the following discussion on page 14.

“Distinct conformations representing the closed, intermediate, and open states and a similar distribution of these conformational states were obtained in both the Na⁺ and K⁺ conditions in cryo-EM experiments. This contrasts with channel activities observed in excised membrane patches, in which the mutant channel, G924S, remained closed in the absence of applied pressure to the excised patches and displayed a lower gating pressure threshold in K⁺, than Na⁺, recording conditions. These discrepancies likely originate from detergent micelles, which are necessary to isolate channel protein for cryo-EM but drastically differ from the excised cell membranes in electrophysiology experiments. Notably, the detergent environment, devoid of lipids, appears to additionally modify the gating properties and does not necessarily recapitulate channel properties in the lipid bilayer, as has been observed in MscS²⁴. In this work, detergent-solubilized MscK channels, in the absence of tension, have provided unanticipated insights into channel gating and allowed us to infer the mechanotransduction process in an ion channel.”

Point 2: Similar to this line of thinking by ephys recordings the authors find that the lower gating threshold for G924S occurs only in KCl conditions but state that their cryo-EM datasets, including relative particle distributions, for NaCl and KCl are identical. It seems that detergent vs lipid has very significant effects on the gating properties and I think it's important for the authors to at least mention this (it may be related to lipid interactions with the protein in the large unoccupied spaces between protein subunits they observe in their structure).

Response: Again, this is an excellent point. Detergent conditions have very significant effects on channel gating and do not necessary recapitulate channel properties in lipid bilayers. Please see the above **Response to Point 1**. We have now included the above discussion in the main text.

Point 3: Although only speculation I would be more careful about any correlation of the two gates pertaining to distinct gating mechanisms. These gates are not "distant" (they are 7

residues apart), and they are part of the same helix that seems to rotate as one during the opening conformational change observed by the authors.

Response: We agree with the reviewer and have now removed the speculation of the correlation of the two gates pertaining to distinct gating mechanisms on page 9.

“It is inviting to correlate these two distant gates with dual activation of MscK, in which each gate is primarily controlled by one of the two stimuli, membrane tension or periplasmic ligand such as K^+ .”

Point 4: I do not agree, at least from what I see, that W914A destabilizes both the closed and open conformations. It is hard to say from a few representative ephys traces but it appears to me that the closed dwell time is similar for W914A and the WT channel and only the open dwell time is markedly decreased. Proper single-channel analysis, which should be simple for these types of recordings, would settle this, otherwise I would be more careful with phrasing. On this point it would be worth commenting more on the interactions of W914 and the other hydrophobic gating residues in the open state. Presumably in the closed state they shield one another from water, forming a closed gate, and in the open state are stabilized by different hydrophobic interactions. In this way one could imagine how the W914A mutation would lead to loss of an interaction in the open state, destabilizing this conformation and reducing open dwell time as appears in the ephys traces. I would include some depiction of these interactions in the extended data figures - perhaps in place of extended data figure 8c which to me is convoluted and hard to interpret.

Response: We appreciate this excellent point. We have now focused on the point that W914A may destabilize the open conformation by decreased hydrophobic interactions in the open state on page 10. We have now included depiction of these interactions in the new Extended Data Fig. 8c.

“It appears that elimination of the bulky side chain destabilizes the open conformation and that W914 is indeed involved in channel gating.”

new Extended Data Fig. 8c, W914 undergoes a significant conformational change during the gating transition from the closed (left panel) to open (right panel) state, which would be hindered by steric obstacles. The protein surface is colored by the lipophilicity potential, with yellow and blue representing hydrophobic and hydrophilic surface, respectively.

Point 5: Although I appreciate the inclusion of all conformations obtained by their cryo-EM analysis it is unclear to me what an "intermediate" conformation of the channel is. Although

limited by resolution the pore-lining helices appear further from one another than in the closed conformation, do the authors expect this pore to be conductive? Are there sub-conductance states for MscK?

Response: The reviewer is correct that limited resolution precluded detailed analysis of the pore dimension. However, sub-conducting states for MscK have been reported in the literature (Li et al, EMBO J 2002; Petrov et al, Channels 2012). In our studies, analysis of MscK substates is complicated by the presence of endogenous MscM activity. MscM has lower unitary conductance and lower gating threshold than MscK. Therefore, MscM activity is almost always present in the recorded traces. We have analyzed a limited number of traces with MscK activity only (or MscK + MscL activities, a total of about 10) and we did not find any long-living MscK substates, though some short-living sub-conducting states might still be present. Unfortunately, our temporal resolution is not sufficient to reliably distinguish between fast complete closures of the channel and its (probable) substates. Additionally, because we cannot have detailed analysis the pore dimension to assign its functional state, we decide not to speculate on the functional state of the intermediate conformation.

Point 6: It is clear that MscK gates at lower membrane tensions than MscL in spheroblast ephys recordings. It is tempting to speculate that this is due, at least partly, to the additional non-planar helices and subsequent membrane deformation of the channel. The notion of such a membrane "footprint" and its contribution to tension sensitivity has been well worked out (see Haselwandter and Mackinnon, <https://doi.org/10.7554/eLife.41968>). I think the discussion would benefit from some elaboration on this point. To me at least it is remarkable to see how nature utilizes local deformation of the bilayer to amplify tension sensitivity in different ion channels and is one of the standout points of the study.

Response: We appreciate this excellent point and have now included discussion on the extent of membrane deformation and tension sensitivity. As also requested by Reviewers 1 and 2, we have now also included estimation of tension sensitivity originated from membrane deformation, which indeed appears to amplify tension sensitivity, analogous to Piezo channels. Please see **Response to Point 1 of Reviewer 1**. We have now included the following discussion on page 13.

"The difference of free energy of midplane bending between the closed (in-plane radius R_c , midplane bending angle θ_c) and open (in-plane radius R_o , midplane bending angle θ_o) states can be approximated by

$$\Delta G_{bending} = \pi(\theta_c^2 R_c - \theta_o^2 R_o) \sqrt{\sigma K_b} \quad (2),$$

where K_b is the bilayer bending modulus $\sim 20 k_B T$ (k_B is Boltzmann's constant, and T is the temperature)⁵⁰. The contribution of membrane deformation to tension sensitivity can be estimated by the midpoint membrane tension ($\sigma_{1/2}$), at which the open probability of the channel is 0.5. Without consideration of $\Delta G_{\sigma=0}$, $\sigma_{1/2}$ is determined by

$$\pi(\theta_c^2 R_c - \theta_o^2 R_o) \sqrt{\sigma_{1/2} K_b} - \sigma_{1/2} \Delta A = 0 \quad (3).$$

Thus, the midpoint tension is calculated by

$$\sigma_{1/2} = \frac{(\theta_c^2 R_c - \theta_o^2 R_o)^2 K_b}{4R_c^2 \Delta R^2} \quad (4).$$

In MscK, the parameters deduced from the closed and open structures result in an estimated midpoint membrane tension of $0.16 k_B T/nm^2$, much lower than the reported midpoint tension of MscL ($2.5 k_B T/nm^2$)⁵¹, which is primarily determined by energetics of hydrophobic mismatch, rather than midplane bending, between the TM helices and the surrounding lipid

bilayer. Opening of MscK is essentially achieved by rigid body movement of the TMD, which appears to maintain a constant membrane thickness matching that of the bacterial inner membrane. Thus, hydrophobic mismatch may not contribute substantially to tension sensitivity of MscK. This analysis implies that MscK could display much higher tension sensitivity than MscL. Analogously, in Piezo channels, the extendedly curved TMD induces a curved 'membrane footprint' that amplifies tension sensitivity⁵². In our analysis, contribution of the intrinsic free energy difference between the closed and open conformations, $\Delta G_{\sigma=0}$, is not included. Notably, the closed and open MscK structures suggest that additional free energy cost must be paid to expand the periplasmic gating ring, thus a higher midpoint membrane tension would be expected."

Reviewer #4 (Remarks to the Author):

In this work, Mount et al determined the cryo-EM structures of a bacterial mechanosensitive channel, MscK, in closed, open, and intermediate states. Structural comparison clearly showed the remarkable conformational changes between closed and open states, including the increase of the heptamer diameter and the decrease of the mid-plane bending angle. Based on the structural observations, they proposed a mechanotransduction mechanism for the mechanosensitive channels. The authors captured three distinct conformations of MscK that strongly support their conclusions. There is no concern or question for the structural part. Several minor suggestions from this referee may help to improve the manuscript:

Point 1: In Figure 2, an additional panel showing how twisted TM11 forms the pore may benefit the readers who don't work in the ion channel field.

Response: We appreciate this good point and have now included a new Extended Data Fig. 9. showing the kinked TM11 forming the ion pore.

Extended Data Fig. 9 | G924S and helix packing. a,b, Packing of pore-lining helices in the closed (a) and open (b) states. Glycine and alanine residues in the pore-lining helices are highlighted as spheres. Positions of the GOF mutations (G922 and G924) are labeled.

Point 2: In Figure 3c and e, please use a bright color to show W914, V921 and F925.

Response: We have now highlighted these residues in green in Fig. 3c and e.

new Fig. 3c-e

Point 3: In Extended Data Table 1, please edit the “PDB xxxx”.

Response: We thank the reviewer and have now included the correct PDB ID.

Reviewers' Comments:

Reviewer #1:

Remarks to the Author:

The authors have very nicely addressed all of the concerns raised in my review.

Reviewer #2:

Remarks to the Author:

The authors addressed all my points except the following two:

Point #1. The authors did not calculate the deformation energy associated with hydrophobic mismatch. The fact that the midpoint threshold for mid-plane bending (0.16 kJ/nm^2) in MscK is much lower than MscL (2.5 kJ/nm^2) suggests that even small changes in hydrophobic mismatch associated with gating could govern MscK opening, a mechanism that would parallel MscL. And contrary to the authors claims, the conformational change in MscK is not limited to a rigid body movement. For example, the inner gating helix changes from bent to straight. As stated previously, the authors are welcome to speculate a membrane bending mechanism for MscK opening, but they need to acknowledge in the discussion that they have not calculated the energy associated with hydrophobic mismatch and therefore, they have not ruled out hydrophobic mismatch as a potentially dominating driver of MscK opening.

Point #2. In Fig 6, the authors did not draw the lipid membrane correctly. Please refer to Figure 7b of Ursell et al 2008. In Ursell et al, the midplane (white line in the grey lipids) bends up and intersects the boarder slope (sloped black line) at 90 degrees. The author's figure does not look like Ursell's. Instead, the authors draw the midplane of their membrane bending down, parallel to the boarder slope. This would suggest the hydrophilic head groups and the lipid tails are perpendicular to boarder slope, which would never happen. The lipids pack parallel to the boarder slope as drawn in Figure 7b, that is precisely what gives rise to the midplane bending. The authors should draw the membrane according to known biology.

Reviewer #3:

Remarks to the Author:

The Authors have addressed all my concerns carefully and thoughtfully and I recommend it's acceptance.

Reviewer #4:

Remarks to the Author:

The authors have addressed all my concerns.

REVIEWER COMMENTS

Reviewer #2 (Remarks to the Author):

Point #1: The authors did not calculate the deformation energy associated with hydrophobic mismatch. The fact that the midpoint threshold for mid-plane bending (0.16 kB/nm²) in MscK is much lower than MscL (2.5 kB/nm²) suggests that even small changes in hydrophobic mismatch associated with gating could govern MscK opening, a mechanism that would parallel MscL. And contrary to the authors claims, the conformational change in MscK is not limited to a rigid body movement. For example, the inner gating helix changes from bent to straight. As stated previously, the authors are welcome to speculate a membrane bending mechanism for MscK opening, but they need to acknowledge in the discussion that they have not calculated the energy associated with hydrophobic mismatch and therefore, they have not ruled out hydrophobic mismatch as a potentially dominating driver of MscK opening.

Response: We acknowledge this point and have now explicitly stated that we have not ruled out hydrophobic mismatch as a potentially dominating driver of MscK opening. We now state the following on page 14.

“...*contribution of hydrophobic mismatch, which may be involved in channel gating, is not included in our estimation of tension sensitivity of MscK.*”

Point #2: In Fig 6, the authors did not draw the lipid membrane correctly. Please refer to Figure 7b of Ursell et al 2008. In Ursell et al, the midplane (white line in the grey lipids) bends up and intersects the boarder slope (sloped black line) at 90 degrees. The author's figure does not look like Ursell's. Instead, the authors draw the midplane of their membrane bending down, parallel to the boarder slope. This would suggest the hydrophilic head groups and the lipid tails are perpendicular to boarder slope, which would never happen. The lipids pack parallel to the boarder slope as drawn in Figure 7b, that is precisely what gives rise to the midplane bending. The authors should draw the membrane according to known biology.

Response: We verify that the bilayer has been drawn correctly in Fig. 6, which is consistent with the depictions of the curved bilayer around analogously curved Piezo channels (see illustrations from references *Nature* 2019 and press release of 2021 Nobel Prize in medicine).

Fig. 1 | Proposed activation mechanisms of PIEZO1. a, Lateral membrane tension model. Changes in membrane properties (for example, tension or curvature) lead to a gating force applied onto PIEZO1. b, Tethered spring model. The PIEZO1 channel is activated through interactions with the cytoskeleton or the extracellular matrix. CED, C-terminal extracellular domain.

Lin YC et al. Force-induced conformational changes in PIEZO1. *Nature* 573:230-234. doi: 10.1038/s41586-019-2004-2 press-

[Redacted] <https://www.nobelprize.org/prizes/medicine/release/>